# Plasma Ag-Modified α-Fe_2_O_3_/g-C_3_N_4_ Self-Assembled S-Scheme Heterojunctions with Enhanced Photothermal-Photocatalytic-Fenton Performances

**DOI:** 10.3390/nano12234212

**Published:** 2022-11-27

**Authors:** Yawei Xiao, Bo Yao, Zhezhe Wang, Ting Chen, Xuechun Xiao, Yude Wang

**Affiliations:** 1National Center for International Research on Photoelectric and Energy Materials, School of Materials and Energy, Yunnan University, Kunming 650504, China; 2Institute of Materials Science & Devices, School of Materials Science and Engineering, Suzhou University of Science and Technology, Suzhou 215009, China; 3Yunnan Key Laboratory of Carbon Neutrality and Green Low-Carbon Technologies, Yunnan University, Kunming 650504, China

**Keywords:** S-Scheme heterojunction, photothermal effect, 2D/2D structure, surface plasmon resonance, photocatalytic-Fenton

## Abstract

Low spectral utilization and charge carrier compounding limit the application of photocatalysis in energy conversion and environmental purification, and the rational construction of heterojunction is a promising strategy to break this bottleneck. Herein, we prepared surface-engineered plasma Ag-modified α-Fe_2_O_3_/g-C_3_N_4_ S-Scheme heterojunction photothermal catalysts by electrostatic self-assembly and light deposition strategy. The local surface plasmon resonance effect induced by Ag nanoparticles broadens the spectral response region and produces significant photothermal effects. The temperature of Ag/α-Fe_2_O_3_/g-C_3_N_4_ powder is increased to 173 °C with irradiation for 90 s, ~3.2 times higher than that of the original g-C_3_N_4_. The formation of 2D/2D structured S-Scheme heterojunction promotes rapid electron-hole transfer and spatial separation. Ternary heterojunction construction leads to significant enhancement of photocatalytic performance of Ag/α-Fe_2_O_3_/g-C_3_N_4_, the H_2_ photocatalytic generation rate up to 3125.62 µmol g^−1^ h^−1^, which is eight times higher than original g-C_3_N_4_, and the photocatalytic degradation rate of tetracycline to reach 93.6%. This thermally assisted photocatalysis strategy improves the spectral utilization of conventional photocatalytic processes and provides new ideas for the practical application of photocatalysis in energy conversion and environmental purification.

## 1. Introduction

Energy crisis and environmental pollution have become urgent problems to be solved, and photocatalytic technology is a green way to tackle them [1,2,3,4]. The core issue of photocatalysis is to increase the activity of the photocatalysts, and many semiconductor materials have been used in photocatalysis so far, such as TiO_2_, CdS, MoSe_2_, g-C_3_N_4_, and perovskite-type composite oxide. Among them, g-C_3_N_4_ is a novel nonmetallic polymer semiconductor, which has been widely used in photocatalysis because of its suitable forbidden bandwidth and tunable energy band structure [5,6,7,8]. The architecture of g-C_3_N_4_ is similar to that of graphene, and it has excellent thermal and chemical stability, which is attributed to the aromatic conjugated system formed by C-N heterocycles and van der Waals forces of interlayer interaction [9,10]. However, for a single g-C_3_N_4_ photocatalyst, the high exciton binding energy makes the photo-generated electrons and holes generated by photoexcitation easily complexed, and the oxidation capacity of the holes on its valence band is not sufficient to generate strongly oxidizing hydroxyl radicals, which is extremely unfavorable for photocatalytic reactions [11,12]. A new S-Scheme heterojunction was proposed to provide a new direction for g-C_3_N_4_ modification recently. The selection of suitable semiconductors to construct S-Scheme heterojunctions with g-C_3_N_4_ can form higher conduction band (CB) positions or lower valence band (VB) positions to enhance the redox ability of g-C_3_N_4_. In addition, the hybridization will generate a new electronic structure, and the potential difference between the two sides of the heterojunction will form an internal electric field and energy band bending at the interface, which promotes photogenerated carrier migration and separation [13,14,15].

The pure g-C_3_N_4_ can only rely on its own photo-excited holes and oxygen-generated radicals generated by photogenerated electrons combined with oxygen to achieve the degradation of pollutants, which is not effective in the degradation of persistent organic pollutants [16,17,18,19,20]. The non-homogeneous Fenton reaction shows a high capacity for the removal of hard-to-degrade organic pollutants, and the photocatalytic-coupled Fenton system is one of the most promising approaches to address water pollution [21,22,23,24,25]. As a typical Fenton catalyst, α-Fe_2_O_3_ has been widely used in photocatalysis because of its suitable band gap, cost-effectiveness, and thermodynamic stability. Combining α-Fe_2_O_3_ with g-C_3_N_4_ to construct S-Scheme heterojunction photocatalytic-Fenton coupling system is a promising scheme to improve the photocatalytic activity of g-C_3_N_4_ [26,27]. The two-electron oxygen reduction reaction (ORR) in the photocatalytic system is capable of generating H_2_O_2_ without additional H_2_O_2_, and the photogenerated electrons capture dissolved oxygen in water and generate H_2_O_2_ near the catalyst, which in turn promotes the following chain reaction between Fe^3+^ and H_2_O_2_ to increase the concentration of hydroxyl radical (·OH) in the system:(1)$$Fe^{2+}+H_{2}O_{2}\to Fe^{3+}+\cdot OH+OH^{-}$$
(2)$$Fe^{3+}+H_{2}O_{2}\to Fe^{2+}+\cdot OOH+H^{+}$$

Moreover, as a narrow bandgap semiconductor, α-Fe_2_O_3_ can effectively broaden the spectral absorption interval of g-C_3_N_4_, improve the spectral utilization of sunlight, and enhance the photothermal conversion effect of the catalyst [28,29]. Therefore, the photocatalytic-Fenton coupling system composed of α-Fe_2_O_3_/g-C_3_N_4_ composites can further improve its photocatalytic performance in theory [30,31].

As a promising photocatalytic technique, surface plasmon resonance has received great attention in recent years [32,33,34,35]. Plasma photocatalysis can extend the absorption range of light through localized surface plasmon resonance (LSPR) [36,37,38]. LSPR of noble metal nanoparticles creates an enhanced surrounding local electric field that can linearly increase the concentration of photogenerated carriers. The effect of the enhanced electric field can also extend to the space charge layer of the adjacent semiconductors, increasing the concentration of photogenerated carriers near the semiconductor surface [39,40]. In addition, the Schottky barrier between the metal nanoparticles and the semiconductor can promote the transfer of the charge carriers to opposite directions, further injecting energetic (hot) electrons into the semiconductor, causing a much higher concentration of energetic electrons on the surface of the photocatalyst [41,42,43]. The energy of the oscillating electrons excited by plasma resonance can be decayed by non-radiative pathways, and the high-energy electrons are coupled to the phonon mode heating the metal lattice through electron-phonon scattering. This heat is eventually diffused into the environment by phonon-phonon relaxation, creating a local thermal effect at the catalyst interface, which speeds up the reaction rate on the catalyst surface [44,45]. In addition, the local electric field generated by LSPR of noble metals facilitates the polarized adsorption of reactant molecules on the surface of the photocatalytic material, which promotes the catalytic activity of the catalyst. Ag has been widely used in photocatalysis because of its cost advantage, abundant reserves, stable nature, and unique antibacterial properties. For instance, Xing et al. introduced silver nanoparticles into the catalytic systems of titanium dioxide and molybdenum sulfide, which showed enhanced light absorption and photocatalytic activity [46,47]. Therefore, the introduction of silver nanoparticles is the preferred way to construct efficient photocatalytic materials.

In this work, Ag/α-Fe_2_O_3_/g-C_3_N_4_ ternary composite photocatalysts were prepared by electrostatic self-assembly and light deposition strategy. The construction of S-Scheme heterojunction in Ag/α-Fe_2_O_3_/g-C_3_N_4_ can effectively drive photogenerated carrier separation and transfer. Ag nanoparticles can expand the light absorption range, provide additional active sites, and also produce excellent photothermal effects through LSPR. The prepared Ag/α-Fe_2_O_3_/g-C_3_N_4_ shows much higher performance than pristine g-C_3_N_4_ for hydrogen production and degradation of tetracycline (TC). The special 2D/2D combination and charge carriers transfer pathway of Ag/α-Fe_2_O_3_/g-C_3_N_4_ were verified by detailed characterization, and the enhanced photothermal effect of Ag/α-Fe_2_O_3_/g-C_3_N_4_ ternary photocatalysts in both gas-phase and liquid-phase was also demonstrated by photothermal testing, then we proposed a synergistic photothermal-photocatalytic-Fenton catalytic mechanism. This work provides a new option for the design of cost-effective, highly active catalysts, as well as a potential way to solve the growing energy crisis and environmental pollution problems.

## 2. Experimental Section

The preparation process of Ag/α-Fe_2_O_3_/g-C_3_N_4_ ternary heterojunction photocatalyst is shown in Figure 1. First, the supramolecular precursors composed of melamine and cyanuric acid were calcined to obtain g-C_3_N_4_ ultrathin nanosheets, and hexagonal α-Fe_2_O_3_ nanosheets were prepared by one step hydrothermally. The zeta potentials of g-C_3_N_4_ and α-Fe_2_O_3_ were measured to be −23.9 mV and +28.3 mV, respectively (Appendix A). g-C_3_N_4_ and α-Fe_2_O_3_ were assembled into heterojunctions driven by electrostatic forces between positive and negative charges, and then Ag/α-Fe_2_O_3_/g-C_3_N_4_ was obtained by modifying Ag nanoparticles on the α-Fe_2_O_3_/g-C_3_N_4_ surface via light deposition process. The specific experimental parameters and test procedures are detailed in the Appendix A.

### 2.1. Preparation of g-C_3_N_4_ ultra-Thin Nanosheets

The bottom-up supramolecular self-assembly method was used to prepare g-C_3_N_4_ ultrathin nanosheets. In a typical synthesis process, 0.01 mol melamine and 0.01 mol cyanuric acid were dissolved in 50 mL of deionized water and stirred at room temperature for 12 h after being mixed. Subsequently, the white melamine-melamine supramolecules were centrifuged and freeze-dried for 24 h. Then, they were warmed up to 550 °C in a muffle furnace at a rate of 5 °C min^−1^ and kept for 4 h. The light-yellow 3D g-C_3_N_4_ was gained after cooling.

### 2.2. Preparation of Hexagonal α-Fe_2_O_3_ Nanosheets

Hexagonal α-Fe_2_O_3_ nanosheets were prepared by one-step solvothermal synthesis. In a typical synthesis, 2.0 mmol of FeCl_3_·6H_2_O was dissolved into a mixed solution of 20 mL ethanol and 1.4 mL water under vigorous magnetic stirring, and 1.6 g of sodium acetate was added after complete dissolution of FeCl_3_·6H_2_O and stirring was continued for 0.5 h. Subsequently, the homogeneous solution was transferred to a 50 mL Teflon-lined reactor sealed with a stainless-steel sheath and kept at 180 °C for 12 h. After cooling to room temperature, the resulting precipitate was rinsed several times with alternating deionized water and ethanol and dried overnight at 60 °C to obtain α-Fe_2_O_3_ hexagonal nanosheets.

### 2.3. Preparation of α-Fe_2_O_3_/g-C_3_N_4_ Heterojunction Photocatalyst

α-Fe_2_O_3_/g-C_3_N_4_ heterojunction photocatalysts were prepared by electrostatic self-assembly method. A total of 50 mg of g-C_3_N_4_ nanosheets were dispersed in 80 mL of deionized water. Then different amounts of α-Fe_2_O_3_ powder were taken and ultrasonically dispersed in 50 mL of deionized water, and the obtained α-Fe_2_O_3_ solution was added to the g-C_3_N_4_ nanosheet suspension under vigorous stirring, and a uniform suspension was obtained after slow stirring for 6 h. After the suspension is filtered, it is washed several times with deionized water and methanol. The obtained mixed powder was dried overnight in an oven at 60 °C to obtain α-Fe_2_O_3_/g-C_3_N_4_ heterojunction photocatalyst.

### 2.4. Preparation of Ag/α-Fe_2_O_3_/g-C_3_N_4_ Ternary Composite Photocatalyst

Ag quantum dots were deposited on the surface of α-Fe_2_O_3_/g-C_3_N_4_ using light deposition process. About 0.2 g of α-Fe_2_O_3_/g-C_3_N_4_ was added to 100 mL of deionized water and stirred for 30 min, then different volumes of AgNO_3_ (0.1 M) solution were added and the mixed solution was irradiated with a 300 W xenon lamp for 30 min under stirring. Finally, the mixed solution was filtered, washed several times with deionized water, and dried in an oven at 50 °C to obtain the Ag/α-Fe_2_O_3_/g-C_3_N_4_ ternary product.

### 2.5. Experiment Characterizations

The X-ray diffractometer (XRD, Rigaku, Japan, λ = 1.5418 Å) was used to determine the phase structure of the prepared photocatalysts. The transmission electron microscope (TEM, JEM-2100, Japan) and scanning electron microscope (SEM, Nova nano SEM 450) were used to examine the microstructure of the photocatalysts, and the distribution of the elements was detected by TEM equipped with the energy dispersive spectrometer (EDS). X-ray photoelectron spectroscopy (XPS) was performed by a photoelectron spectrometer (Thermo ESCALAB 250Xi, USA). The ultraviolet-visible (UV-Vis) diffuse reflectance spectra of the catalysts were measured by a spectrophotometer (UV-2600i, Shimadzu, Japan). Photoluminescence (PL) spectra of the photocatalysts were measured with an FLS 1000 fluorescence spectrophotometer (Edinburgh Instruments). The work function of samples was tested by Scanning Kelvin Probe (SKP) (SKP5050 system, Scotland).

### 2.6. Photothermal Test

The photothermal test of as-prepared samples was carried out as follows. About 0.1 g of sample was laid out flat on white weighing paper (Appendix A) and the initial temperature was controlled at room temperature. The temperature of the sample was measured using the Testo 865 infrared thermography. In the water temperature evolution test, 0.05 g photocatalyst was dispersed in 30 mL of water and kept magnetically stirred, and the corresponding temperature was recorded every 10 min by infrared thermal imager Testo 865. A 300 W xenon lamp (CEL-HXF300, Beijing China Education Au-light Co., Ltd., Beijing, China) was used as the light source for all the photothermal experiments, with a distance of approximately 20 cm between the light source and the sample.

### 2.7. Photocatalytic Performance Assessment

The photocatalytic pollutant degradation reactions were carried out on a multi-purpose photochemical reaction system (CEL-LAB500E4, Beijing China Education Au-light Co., Ltd.). About 50 mg of the photocatalyst was added to 100 mL of a solution containing TC (10 mg/L). Before the photocatalytic experiments, the solution containing the pollutant and the photocatalyst was placed in a dark room for 30 min to get the adsorption-desorption equilibrium. Then, the solution was irradiated under a 300 W xenon lamp with circulating water jacket (CEL-HXF300, Beijing China Education Au-light Co., Ltd.). Every 20 min, 3 mL of each liquid sample was removed from the beaker and filtered with 0.22 μm Millipore filter heads. Concentrations were subsequently tested using a high-performance liquid chromatograph (LC-3100).

The photocatalytic hydrogen evolution reaction was performed on the all-glass automatic online trace gas analysis system (Labsolar-6A, Beijing Perfect light Technology Co., Ltd., Beijing, China). With Labsolar-6A, a 300 W Xenon lamp (Microsolar300, Beijing Perfect light Technology Co., Ltd.) was used as the simulated sunlight spectral source. The as-prepared catalyst (10 mg) was uniformly dispersed by using a magnetic stirrer in 120 mL of methanol solution (containing H_2_O/methanol, *v*/*v* = 90:30). The temperature of the reaction was kept at 298 K by cool flowing water. During the irradiation process, a hydrogen sample (0.5 mL) was extracted from the reactor at a given interval and the amount of hydrogen produced was analyzed by an online gas chromatograph (GC-7900).

## 3. Results and Discussion

### 3.1. Physical Phase Composition and Energy Band Structure

The crystal structures of g-C_3_N_4_ nanosheets, hexagonal α-Fe_2_O_3_ nanosheets, α-Fe_2_O_3_/g-C_3_N_4_ 2D/2D heterojunction and Ag/α-Fe_2_O_3_/g-C_3_N_4_ ternary heterojunction were analyzed by XRD pattern, as shown in Figure 2a. The g-C_3_N_4_ has two distinct peaks at 12.8° and 27.7°, which were derived from the repeating unit and interlayer stacking within the carbon nitride plane, corresponding to two crystal planes (100) and (002), respectively [48]. The characteristic peaks of α-Fe_2_O_3_ at 24.1, 33.2, 35.7, 40.8, 49.5, 54.1, 62.5, and 64.1° correspond to the crystal faces of (012), (104), (110), (113), (024), (116), (214), and (300) on standard α-Fe_2_O_3_ cards (JCPDS No. 33-0664), respectively. The sharp XRD peaks indicate the high crystallinity of α-Fe_2_O_3_. The characteristic peaks of g-C_3_N_4_ and α-Fe_2_O_3_ are observed in the α-Fe_2_O_3_/g-C_3_N_4_ composite sample, indicating that the α-Fe_2_O_3_/g-C_3_N_4_ heterojunction photocatalyst was successfully prepared. The diffraction peaks of Ag were not observed in the XRD patterns of the Ag/α-Fe_2_O_3_/g-C_3_N_4_ ternary composites due to the low loading of Ag nanoparticles [40,43,47]. However, in the XPS with higher sensitivity, all the characteristic peaks of the α-Fe_2_O_3_ and g-C_3_N_4_ appeared in the XPS survey spectrum of Ag/α-Fe_2_O_3_/g-C_3_N_4_ composites, and the characteristic peaks of Ag were observed near 367 and 373 eV, indicating that the Ag/α-Fe_2_O_3_/g-C_3_N_4_ ternary composites were successfully prepared (Figure 2b) [47,49]. The light absorption properties of each prepared catalytic material were characterized by UV-vis diffuse reflectance spectroscopy (UV-vis DRS), Figure 2c shows that the optical response interval of the α-Fe_2_O_3_/g-C_3_N_4_ expands from 483 nm to 647 nm. The optical response of the Ag/α-Fe_2_O_3_/g-C_3_N_4_ ternary composite interval was further extended to the NIR region due to the LSPR of Ag nanoparticles. The intrinsic band gaps of g-C_3_N_4_ and α-Fe_2_O_3_ were estimated from the Tauc plots in Figure 2d, and the forbidden bandwidths of 2.04 and 2.58 eV for g-C_3_N_4_ and α-Fe_2_O_3_ were calculated from the intercepts of extrapolated lines in the *X*-axis, respectively. The energy band structure of Ag/α-Fe_2_O_3_/g-C_3_N_4_ was determined by VB-XPS. The test results of the VB-XPS in Figure 2e show that the VB positions of α-Fe_2_O_3_ and g-C_3_N_4_ are 2.36 and 1.35 eV relative to the normal hydrogen electrode, respectively. The relative energy band positions of α-Fe_2_O_3_ and g-C_3_N_4_ can be calculated based on the relationship between the semiconductor VB and CB, as shown in Figure 2f, the energy band arrangement of α-Fe_2_O_3_ and g-C_3_N_4_ is similar to Type II heterojunction [50]. The g-C_3_N_4_ possesses a higher conduction band position and is more electronically reductive, and the α-Fe_2_O_3_ has a lower valence band position and strong hole oxidation, which means that α-Fe_2_O_3_ and g-C_3_N_4_ will form S-Scheme heterojunctions with both strong proton reduction and strong hole oxidation ability.

### 3.2. Morphological and Structural Analysis

The morphological and structural characteristics of different photocatalysts were analyzed by TEM and SEM. In Figure 3a, g-C_3_N_4_ exhibits a self-supporting nanosheet morphology with a thickness of about 10 nm, which is further observed by the nearly transparent TEM image (Figure 3d). As shown in Figure 3b,g, the SEM and TEM images of α-Fe_2_O_3_ indicate that it is a hexagonal nanosheet with a size of about 220 nm and a thickness of about 20 nm. The SEM and TEM of α-Fe_2_O_3_/g-C_3_N_4_ composites are shown in Figure 3c,e, respectively, where two differently charged sheets of α-Fe_2_O_3_, g-C_3_N_4_ are closely adhered by electrostatic force. The hexagonal α-Fe_2_O_3_ nanosheets tightly adhere to the substrate composed of g-C_3_N_4_ with large size. This 2D/2D heterojunction shortens the distance that charge carriers have to migrate to the surface and facilitates the rapid involvement of carriers in chemical reactions [7]. Figure 3f clearly shows that Ag nanoparticles are dispersed on the α-Fe_2_O_3_/g-C_3_N_4_ surface. Figure 3h,i is high-resolution TEM images of the circled region, respectively, which show that the fully exposed active crystal plane of α-Fe_2_O_3_ is the (110) plane, while the 0.24 nm lattice spacing corresponds to the (111) crystal plane of Ag [51,52]. Figure 3j–n shows the energy dispersive X-ray spectroscopy (EDX) of Ag/α-Fe_2_O_3_/g-C_3_N_4_, which shows the presence of N (green), C (yellow), O (red), Fe (violet), and Ag (indigo) elements, further demonstrating the successful synthesis of Ag/α-Fe_2_O_3_/g-C_3_N_4_ ternary composite photocatalyst.

### 3.3. Analysis of Charge Transfer Mechanism

The charge transfer upon contact in compound semiconductors is generally related to the work function of the materials. To study the charge movement between g-C_3_N_4_ and α-Fe_2_O_3_, the work functions of them were analyzed by SKP (Figure 4a). The test results are presented as relative values of Au, and the work function of the standard Au sample was 5.1 eV. The work functions of α-Fe_2_O_3_ and g-C_3_N_4_ were derived to be 5.61 and 4.54 eV from the difference between the test results and the work function of the standard Au sample. Apparently, g-C_3_N_4_ has a higher Fermi energy position relative to α-Fe_2_O_3_, and when they are in close contact to form a heterojunction, the difference in work function leads to the spontaneous diffusion of electrons from g-C_3_N_4_ to α-Fe_2_O_3_ and form electron depletion and electron aggregation layers at the contact interface, which is the reason for the formation of the internal electric field. In addition, changes in element valence and electron density lead to chemical shifts in XPS, so the charge transfer mechanism of α-Fe_2_O_3_/g-C_3_N_4_ heterojunction can be analyzed by in situ irradiated XPS. Peaks in the Ag 3d XPS spectrum appear at 373.3 and 367.3 eV, originating from Ag 3d_5/2_ and Ag 3d_3/2_ with a difference of 6.0 eV between the two peaks, indicating that Ag exists as a monomeric form (Figure 4b) [49]. Figure 4c shows that the peaks of C1s located at 286.5 and 288.3 eV are attributed to C-N and C-N=C in g-C_3_N_4_, and the carbon peak at 284.8 eV is used to calibrate the other peaks. The peaks of N1s at 398.5, 399.8, and 400.8 eV in Figure 4d are assigned to C-N=C, C-(N)_3_ and C-N-H in g-C_3_N_4_ [53]. Upon contact between α-Fe_2_O_3_ and g-C_3_N_4_, electrons transfer from g-C_3_N_4_ to α-Fe_2_O_3_ due to the difference in work functions. Both the peaks of C and N move toward higher binding energies after losing electrons. The electrons in the CB of α-Fe_2_O_3_ move to g-C_3_N_4_ driven by the internal electric field under light, and the peaks of N and C move toward lower binding energy after g-C_3_N_4_ gaining electrons. The peaks of Fe 2p appeared at 711.2, 719.0, and 724.3 eV in Figure 4e, corresponding to Fe 2p_3/2_, Fe^3+^, and Fe 2p_1/2_, respectively. Figure 4f shows that the characteristic peaks of O1s are located at 529.7 and 531.4 eV, attributing to lattice oxygen and adsorbed oxygen in α-Fe_2_O_3_, respectively, and the peak at 532.8 eV is from water adsorbed by the composite photocatalyst [54,55]. The shifting trend of O and Fe in α-Fe_2_O_3_/g-C_3_N_4_ before and after light illuminating is opposite to that of g-C_3_N_4_. α-Fe_2_O_3_ gets electrons when a heterojunction is formed, and its binding energy decreased. However, after illumination, α-Fe_2_O_3_ loses electrons and its binding energy rises [13,14,19]. Kelvin probe and in situ irradiated XPS results demonstrate the charge transfer mode of α-Fe_2_O_3_/g-C_3_N_4_ S-Scheme heterojunction, as shown in Figure 4g. Electrons move from g-C_3_N_4_ to α-Fe_2_O_3_ due to the difference of work function between α-Fe_2_O_3_ and g-C_3_N_4_, and an internal electric field is formed at the interface of the heterojunction, and the energy band bending of α-Fe_2_O_3_ and g-C_3_N_4_ caused by the electrostatic repulsion. When the α-Fe_2_O_3_/g-C_3_N_4_ heterojunction photocatalyst is excited by light, the electrons in the CB of α-Fe_2_O_3_ move to the VB of g-C_3_N_4_ and compound with the holes under the effect of internal electric field and energy band bending, so that the α-Fe_2_O_3_/g-C_3_N_4_ heterojunction maintains strong redox properties.

### 3.4. Photothermal Effect Evaluation

The photothermal effect of the catalyst was characterized by infrared thermography. Figure 5a,b shows the change of surface temperature of the pure g-C_3_N_4_ and ternary heterojunction photocatalysts with the increased light time in the air, respectively. The initial temperature of the g-C_3_N_4_ powder surface was 26.6 °C, and then increased quickly to 54.9 °C after 90 s of light illumination, while the temperature of Ag/α-Fe_2_O_3_/g-C_3_N_4_ powder increased from 26.5 °C to 173 °C at the same condition, which was 5.57 times higher than that of the original g-C_3_N_4_, indicating that the construction of heterojunction and the modification of Ag nanoparticles significantly improved the photothermal conversion ability of the ternary composite. The photothermal performance of the powder catalysts in real reactions was investigated by photothermal experiments in water. Figure 5c,d shows the temperature changes of pristine g-C_3_N_4_ and the ternary composite sample with the increase of light time in the water, respectively. The temperature of g-C_3_N_4_ solution system under light increased from the initial 19.4 °C to 32.3 °C with a temperature increase of 12.9 °C, while the temperature of the ternary heterojunction increased from the initial 19.7 °C to 59.1 °C at the same condition, which was three times higher than that of pure g-C_3_N_4_, indicating that Ag/α-Fe_2_O_3_/g- C_3_N_4_ offers a more significant photothermal effect. Ag/α-Fe_2_O_3_/g-C_3_N_4_ ternary photocatalysts can convert part of the absorbed photon energy into thermal energy through the photothermal effect to form a local thermal environment near the catalyst surface in a photocatalytic process, thus increasing the near-field chemical reaction rate of the catalyst and enhancing the photocatalytic performance of the catalyst [13,56,57].

### 3.5. Photocatalytic Hydrogen Production and Pollutant Degradation Testing

Photocatalytic hydrogen production and degradation of TC experiments were performed to evaluate the activity of prepared catalysts. The synthesis process was optimized by degrading TC experiment. The photocatalytic performance of α-Fe_2_O_3_/g-C_3_N_4_ heterojunction was affected by the ratio of α-Fe_2_O_3_. The best degradation effect of TC was obtained when the weight ratio of α-Fe_2_O_3_ was 30wt%. The α-Fe_2_O_3_/g-C_3_N_4_ will produce serious agglomeration if α-Fe_2_O_3_ is too much (Appendix A), and the performance improvement is not obvious if α-Fe_2_O_3_ is too low (Appendix A). When the loading amount of Ag nanoparticles is 0.5wt%, Ag/α-Fe_2_O_3_/g-C_3_N_4_ has the best photocatalytic performance (Appendix A). The performance of the catalyst is not improved as the silver nitrate increased, indicating that the content of Ag particles loaded on α-Fe_2_O_3_/g-C_3_N_4_ by photoreduction process was relatively low, which is beneficial to improve the dispersion of Ag nanoparticles. Therefore, samples with optimal composite ratios were used to assess the contribution of constructing heterojunctions and plasma Ag modifications. Figure 6a shows that the original g-C_3_N_4_ photocatalytic hydrogen production rate was 387.45 µmol g^−1^ h^−1^, and the hydrogen production rate of ternary heterojunction reached 3125.62 µmol g^−1^ h^−1^. After constructing the S-Scheme heterojunction and completing the surface-engineered plasma Ag modification, the hydrogen production rate reached more than eight times of the original one, and there was no significant decay during the 1200 min cycle (Figure 6b). Ag/α-Fe_2_O_3_/g-C_3_N_4_ has a high hydrogen production efficiency compared to other recently reported photocatalysts of the same type (Appendix A). Ag nanoparticles not only improve carrier separation through LSPR but also provide additional active sites for hydrogen production [33,34]. In addition, Ag/α-Fe_2_O_3_/g-C_3_N_4_ also showed the best performance in the degradation experiments of TC with a 93.6% degradation rate in 150 min (Figure 6c), and approximately maintained this degradation rate for all five cycles (Figure 6d). The first-order reaction kinetic curves show that apparent reaction rate of Ag/α-Fe_2_O_3_/g-C_3_N_4_ was 2.6 times that of pure g-C_3_N_4_ (Figure 6e), and Ag/α-Fe_2_O_3_/g-C_3_N_4_ has relatively excellent TC degradation compared to other recently reported photocatalysts of the same type (Appendix A), which was the combined effect of S-Scheme heterojunction and photothermal effect to enhance the chemical reaction rate [19,56]. The radical quenching experiments showed that ·OH played a dominant role in the degradation of TC, followed by vacancies, and the superoxide radical (${\cdot O}_{–2}^{–2}$) at last (Figure 6f) [58]. The mineralization rate of ternary heterojunction for TC reached 49.26%, which was 44.3 times that of pristine g-C_3_N_4_, indicating that the redox ability of the composite sample was substantially enhanced (Figure 6g). To investigate the promotion of photocatalytic degradation reaction by photothermal effect, we conducted a circulating cooling experiment. The effect of temperature on the degradation of TC by Ag/α-Fe_2_O_3_/g-C_3_N_4_ was relatively obvious from Figure 6h, and the degradation efficiency decreased significantly when the circulating cooling water was turned on. The reaction rate in the Ag/α-Fe_2_O_3_/g-C_3_N_4_ degradation of TC without cooling water was 1.5 times higher than that with cooling water (Figure 6i), which indicated that the heat generated by the photothermal effect could effectively promote the photocatalytic catalytic reaction rate.

### 3.6. Photoelectrochemical Performance Analysis

To investigate the charge transport kinetics of the photocatalyst, a series of photoelectrochemical tests was performed. Figure 7a shows that Ag/α-Fe_2_O_3_/g-C_3_N_4_ has the best transient photocurrent response, indicating its high concentration of photogenerated carriers. The low impedance implies a low carrier migration resistance in Ag/α-Fe_2_O_3_/g-C_3_N_4_, which facilitates charge separation (Figure 7b). The steady-state PL shows the lowest luminescence intensity in Ag/α-Fe_2_O_3_/g-C_3_N_4_, demonstrating that the S-Scheme heterojunction effectively suppresses the radiative relaxation in Ag/α-Fe_2_O_3_/g-C_3_N_4_ (Figure 7c). Time-resolved photoluminescence spectroscopy (TRPL) can accurately analyze the carrier lifetime in the catalysts, and the fitted results in Figure 7d show that the fluorescence lifetime of Ag/α-Fe_2_O_3_/g-C_3_N_4_ is 4.57 ns, which is 3.7 times that of the pristine g-C_3_N_4_, indicating that the S-Scheme heterojunction and surface plasmon resonance effects enhance the separation and transport of carriers. The electron paramagnetic resonance (EPR) experiments reveal the mechanism of the interaction between carriers and surface-adsorbed molecules in the catalysts. The ·O_2_^-^ and ·OH production under light for each catalyst is shown in Figure 7e,f, respectively. The results of DMPO-·O showed that α-Fe_2_O_3_ could not generate ${\cdot O}_{2}^{–}$ under light, which is due to its conduction band electron reduction ability is not sufficient to reduce O_2_ to generate ${\cdot O}_{2}^{–}$. The signal of ${\cdot O}_{2}^{–}$ was detected after α-Fe_2_O_3_ was compounded with g-C_3_N_4_, and the Ag nanoparticle modification further enhanced the ${\cdot O}_{2}^{–}$ signal. The DMPO-·OH results showed that no ·OH signal is detected in pure g-C_3_N_4_ due to insufficient oxidation capacity of the hole. α-Fe_2_O_3_/g-C_3_N_4_ and Ag/α-Fe_2_O_3_/g-C_3_N_4_ showed obvious ·OH characteristic signals, which indicated that the composite photocatalytic system can generate a large number of strongly oxidizing hydroxyl radicals under light to participate in the degradation reaction. This is consistent with the results of the free radical quenching experiments. The ERP test results further reveal that charge transfer similar to that of Type II heterojunctions does not occur in the composite photocatalytic system and that electrons and holes with strong redox capabilities are retained in the system. The mode of charge transfer in ternary heterojunctions is consistent with the S-Scheme mechanism [13,14].

### 3.7. Mechanistic Analysis

Based on the above experimental phenomena and characterization results, we propose a photocatalytic mechanism for Ag/α-Fe_2_O_3_/g-C_3_N_4_ heterojunction. S-Scheme heterojunction, photothermal and photocatalytic-Fenton synergy for Ag/α-Fe_2_O_3_/g-C_3_N_4_ performance enhancement (Figure 8). The carrier generated by photoexcitation migrates directionally under the action of internal electric field and interfacial energy band bending, and its migration path follows the S-Scheme mechanism that electrons and holes stored in the VB of α-Fe_2_O_3_ and in the CB of g-C_3_N_4_, respectively. The electron with strong reducibility in g-C_3_N_4_ can capture free hydrogen and dissolved oxygen and generate H_2_O_2_ rapidly. The free electrons in the photocatalytic system can reduce part of Fe^3+^ to Fe^2+^ in α-Fe_2_O_3_, Fe^2+^ then react with H_2_O_2_ to produce a large amount of ·OH through the chain Fenton reaction. Moreover, a large number of high-energy electrons generated by the LSPR effect of Ag nanoparticles are injected into the S-Scheme heterojunction photocatalytic system, which facilitates the generation of strong oxidizing ·OH and ${\cdot O}_{2}^{–}$ and increases the concentration of free radicals in the system, and the heat generated will promote the rate of chemical reactions on the surface. This photocatalytic reaction process consists of the following steps:(3)$$Ag/\alpha –Fe_{2}O_{3}/g–C_{3}N_{4}+hv\to e^{-}+h^{+}$$
(4)$$e^{-}+O_{2}\to \cdot O_{2}{}^{-}$$
(5)$$\cdot O_{2}{}^{-}+H^{+}\to H_{2}O_{2}$$
(6)$$h^{+}/\cdot O_{2}{}^{-}/\cdot OH+pollutants\to H_{2}O+CO_{2}$$

The photocatalytic-Fenton coupling system increases the number of active radicals effectively, and the internal energy generated by the photothermal effect can be activated to promote the dissociation of water molecules and accelerate the chemical reaction kinetics. The synergistic effect of S-Scheme heterojunctions, photothermal and photocatalytic-Fenton enabled the Ag/α-Fe_2_O_3_/g-C_3_N_4_ ternary heterojunction to exhibit satisfactory photocatalytic performance [23,24,25].

## 4. Conclusions

In conclusion, we have successfully prepared plasma Ag-modified α-Fe_2_O_3_/g-C_3_N_4_ S-Scheme heterojunctions by electrostatic self-assembly and light deposition strategy. The photocatalytic hydrogen production rate of Ag/α-Fe_2_O_3_/g-C_3_N_4_ reached 3125 µmol g^−1^ h^−1^, eight times that of pristine g-C_3_N_4_, and the photocatalytic TC degradation rate was as high as 93.6% within 120 min. The reason for this enhanced performance may arise from the following: (1)The S-Scheme heterojunction composed of 2D/2D nanosheets facilitates the spatial separation of charge carriers and the full exposure of active sites; (2) the photocatalytic-Fenton reaction helps to improve the carrier conversion and to enrich the number of active radicals in the reaction system; (3) the LSPR effect of Ag nanoparticles provides high-energy electrons the system to promote the separation of carriers, and generates the photothermal effect to facilitate surface reaction kinetic. This synergistic catalytic strategy provides valuable insights for the construction of highly active photocatalytic systems.

## Figures and Tables

**Figure 1 nanomaterials-12-04212-f001:**
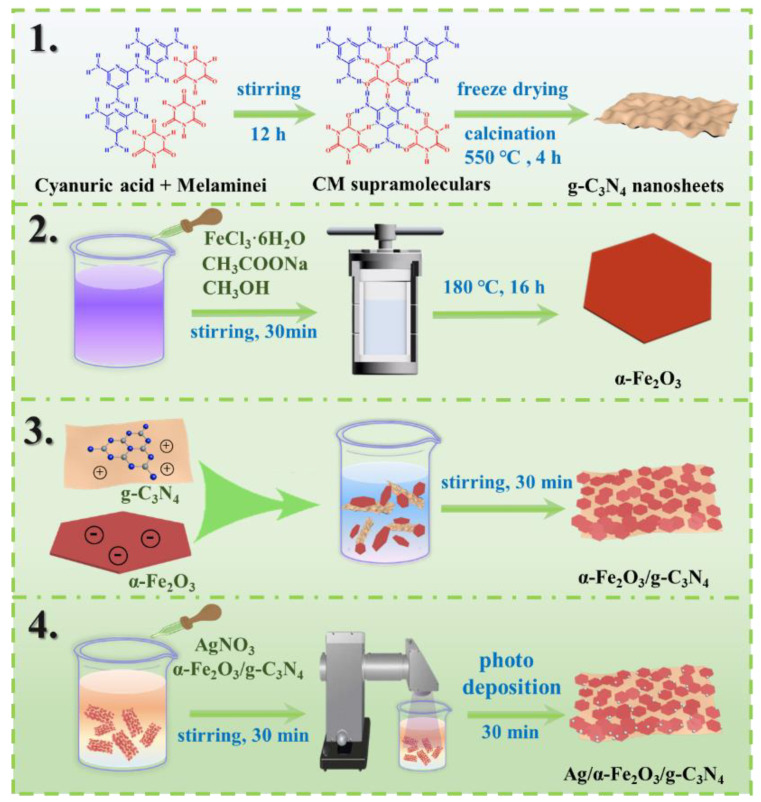
Scheme diagram of the preparation process of Ag/α-Fe_2_O_3_/g-C_3_N_4_ sandwich-like self-assembled S-Scheme heterojunction photocatalyst.

**Figure 2 nanomaterials-12-04212-f002:**
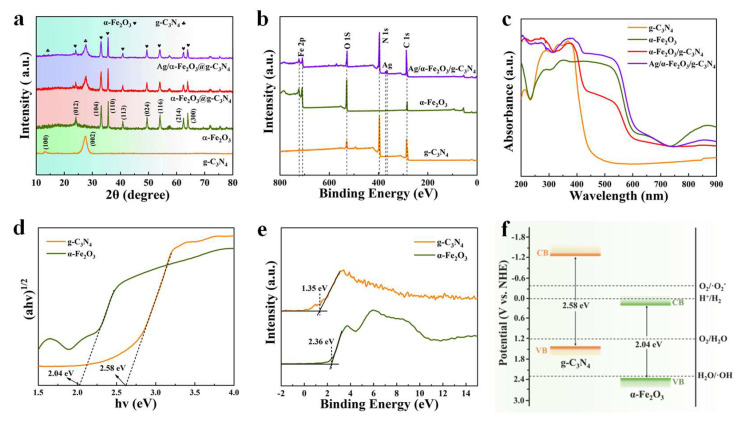
XRD patterns (**a**), XPS full spectra (**b**) and UV-vis DRS spectra (**c**) of g-C_3_N_4_, α-Fe_2_O_3_ and Ag/α-Fe_2_O_3_/g-C_3_N_4_, band gap (**d**), VB-XPS I (**e**) and relative energy band position maps (**f**) of g-C_3_N_4_ and α-Fe_2_O_3_.

**Figure 3 nanomaterials-12-04212-f003:**
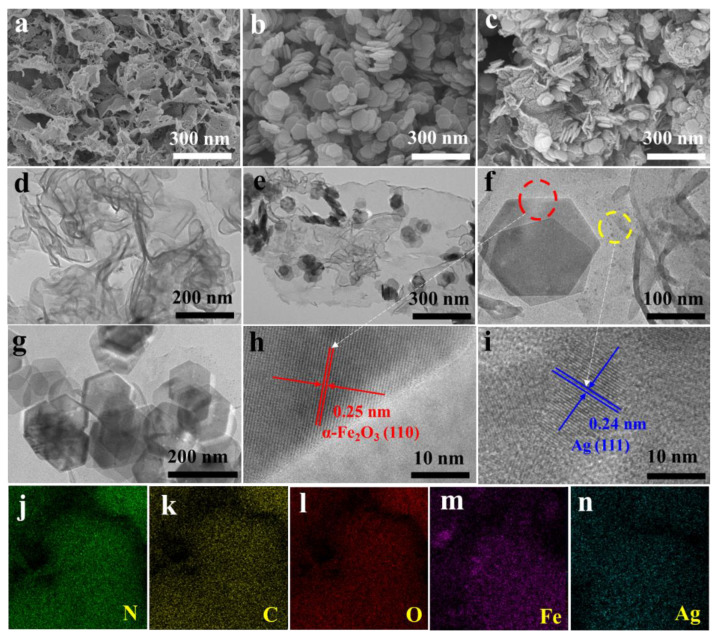
SEM image of (**a**) g-C_3_N_4_, (**b**) α-Fe_2_O_3_ and (**c**) α-Fe_2_O_3_/g-C_3_N_4_, TEM image of (**d**) g-C_3_N_4_, (**e**) α-Fe_2_O_3_/g-C_3_N_4_, (**f**) Ag/α-Fe_2_O_3_/g-C_3_N_4_, (**g**) α-Fe_2_O_3_, HRTEM image of (**h**) α-Fe_2_O_3_ and (**i**) Ag nanoparticles, (**j**–**n**) SEM-EDX mapping of Ag/α-Fe_2_O_3_/g-C_3_N_4_.

**Figure 4 nanomaterials-12-04212-f004:**
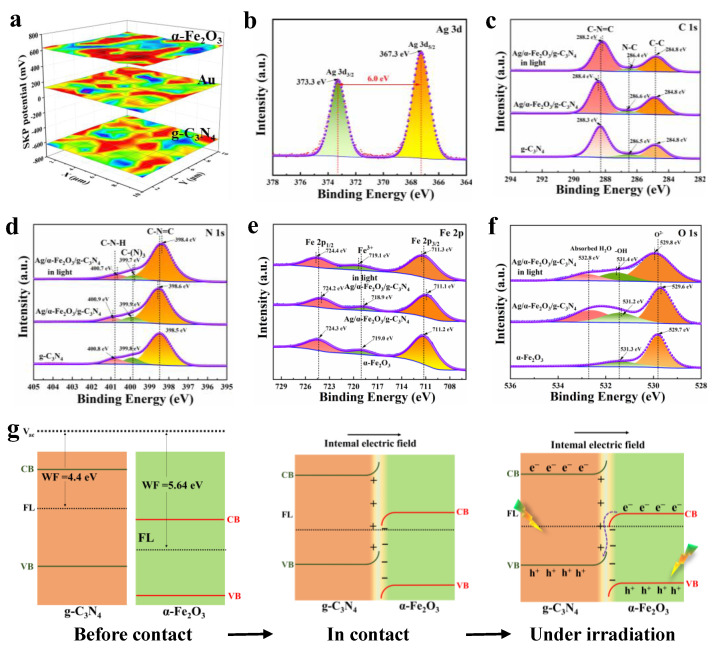
Work function of α-Fe_2_O_3_ and g-C_3_N_4_ (**a**). Ag 3d XPS spectra (**b**), the XPS spectra of C 1s (**c**) and N 1s (**d**) of g-C_3_N_4_ and Ag/α-Fe_2_O_3_/g-C_3_N_4_, the XPS spectra of O 1s (**e**) and Fe 2p (**f**) of α-Fe_2_O_3_ and Ag/α-Fe_2_O_3_/g-C_3_N_4_. (**g**) The formation mechanism of S-Scheme heterojunctions.

**Figure 5 nanomaterials-12-04212-f005:**
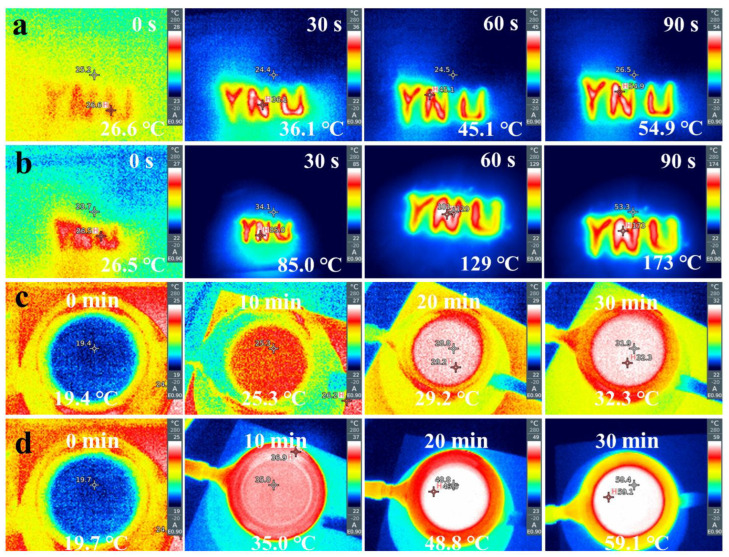
Temperature-light time trends of g-C_3_N_4_ (**a**), Ag/α-Fe_2_O_3_/g-C_3_N_4_ (**b**) in the air. Temperature-light time trends of g-C_3_N_4_ (**c**), Ag/α-Fe_2_O_3_/g-C_3_N_4_ (**d**) in the water.

**Figure 6 nanomaterials-12-04212-f006:**
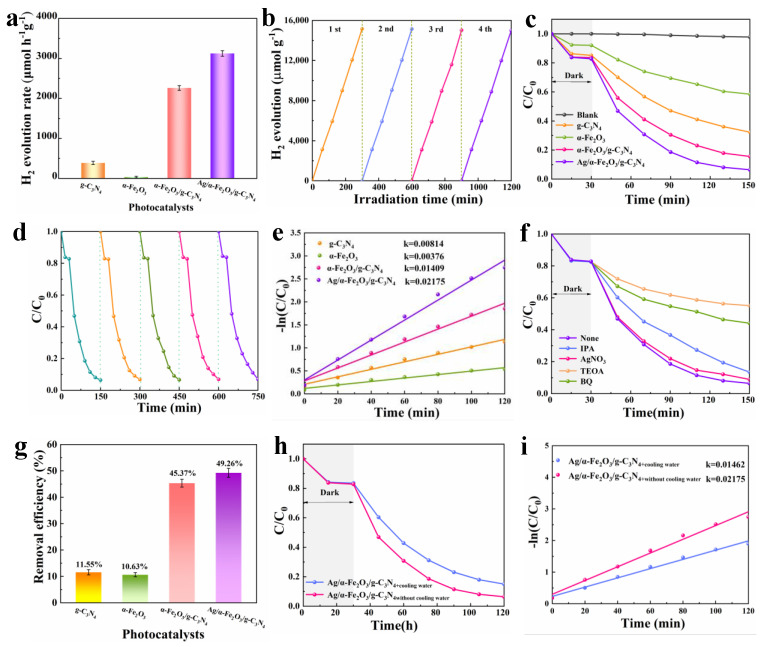
Hydrogen production rate (**a**), hydrogen production rate cycling performance (**b**), photocatalytic-Fenton degradation performance (**c**), degradation cycling performance (**d**), the first-order kinetic curves (**e**), free radical quenching experiments (**f**), contaminant mineralization performance (**g**), cyclic cooling experiments (**h**), cyclic cooling first-order kinetic curves (**i**) for g-C_3_N_4_, α-Fe_2_O_3_, α-Fe_2_O_3_/g-C_3_N_4_ and Ag/α-Fe_2_O_3_/g-C_3_N_4_.

**Figure 7 nanomaterials-12-04212-f007:**
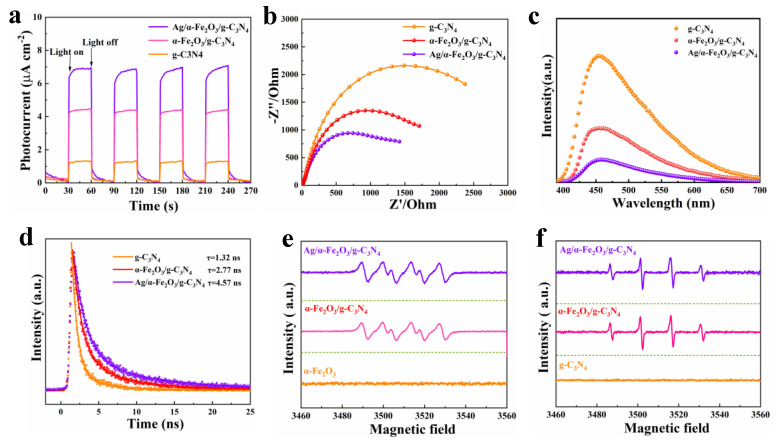
Photocurrent curves (**a**), impedance experiments (**b**), PL spectra (**c**), TRPL spectra (**d**), DMPO-${\cdot O}_{2}^{–}$ signals under light (**e**), DMPO-·OH signals under light (**f**) of g-C_3_N_4_, α-Fe_2_O_3_/g-C_3_N_4_ and Ag/α-Fe_2_O_3_/g-C_3_N_4_.

**Figure 8 nanomaterials-12-04212-f008:**
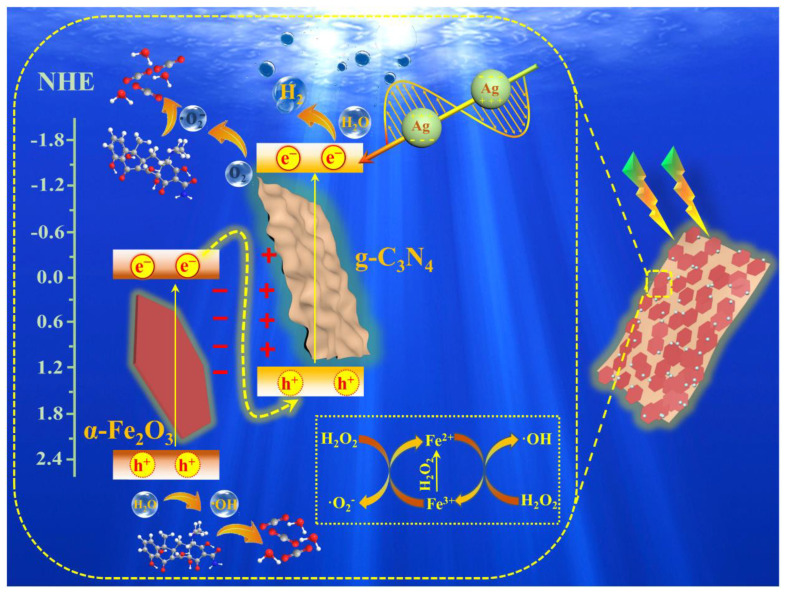
Schematic diagram of photocatalytic mechanism of Ag/α-Fe_2_O_3_/g-C_3_N_4_ S-Scheme heterojunction.

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
