# Peer review of "Plasma Ag-Modified α-Fe2O3/g-C3N4 Self-Assembled S-Scheme Heterojunctions with Enhanced Photothermal-Photocatalytic-Fenton Performances"

_nanomaterials, 2022, doi:10.3390/nano12234212_

Round 1

Reviewer 1 Report

The manuscript describes the development of ternary materials of Ag/a-Fe2O3/carbon nitride that were applied as photocatalysts for hydrogen production and the degradation of pollutants. The manuscript is interesting but there are some important drawbacks. In none of the applications the authors did not study the effect of the temperature. Therefore, for the reviewer, there is no photothermal study. Also, there is no reason to add the term “sandwich-like” in the title. I also found one major issue regarding the tests for hydrogen production. Iron oxide produced some hydrogen. Are the CB electrons on the oxide negative enough to produce hydrogen? Also, the authors did not use any co-catalyst. Is it possible to produce H2 using pure carbon nitride without co-catalyst? I cannot recommend publication. In the following, specific comments are also given.

Specific comments

Details regarding materials characterization and catalytic experiments must be given in the experimental section.

Synthesis part in Supporting info, 2.1. Preparation of g-C3N4 ultra-thin nanosheets: “0.01 mol melamine and 0.01 mol melamine were…” please correct. Is it cyanuric acid? It is also mentioned that the material was thermally treated at 550 degrees but in figure 1 it is 500 degrees. Which one is correct?

P6L1: “and the diffraction peaks” authors discuss the XPS data at this point. Not the XRD. There are no diffraction peaks. Please revise.

P8: “is opposite to that of nitrogen carbide” nitrogen carbide?

Figure 6a: There is a small H2 production from pure a-Fe2O3. Is this possible? What is the CB potential of this material? Also, for the photocatalytic H2 production the authors did not use any co-catalyst. It is rather difficult to believe that there is activity (in H2 photocatalytic production) using pure carbon nitride without any co-catalyst. These two points make it difficult to believe catalytic for H2 production. There is a big problem on this matter.

There is no study on the effect of the content of each part. There is no study of the content of each part in the composites developed. What is the amount of a-Fe2O3 and Ag? The effect of the content of each part must be studied. Optimization must be performed.

There was no temperature control of the degradation reactions but there was a temperature control for the reactions performed to study H2 evolution.

Details must be given in the experimental part. More details for the photothermal tests are required. How was the 0.1g loaded on white paper?

Papers like http://dx.doi.org/10.1016/j.apcatb.2016.01.013 and references therein for the development of composites of carbon nitride with iron oxides and silver nanoparticles must be also included and properly discussed.

Author Response

According to the comments, we have revised the manuscript.

Author Response

(The authors gave the same response as above.)

Reviewer 3 Report

The present manuscript describes the Plasma silver-modified sandwich-like Fe2O3/g-C3N4 self-assembled S-Scheme heterojunctions with enhanced photothermal-photocatalytic Fenton performance. This reviewer recommends the publication after addressing the following comments of this manuscript in Nanomaterials.

In abstract section should contain some quantitative information.

In this work, Ag/α-Fe2O3/g-C3N4 ternary t composite photocatalysts. What is the ternary t composite?

Synthesis section should be moved to main manuscript from the supporting information. (2.1. Preparation of g-C3N4 ultra-thin nanosheets, 2.2 Preparation of hexagonal α-Fe2O3 nanosheets, 2.3 Preparation of α-Fe2O3/g-C3N4 heterojunction photocatalyst, and 2.4 Preparation of Ag/α-Fe2O3/g-C3N4 ternary composite photocatalyst)

The author should cite related references in all the results and discussion section.

There is no any outstanding point of this in conclusions section. The author should revise it.

Author Response

(The authors gave the same response as above.)

Round 2

Reviewer 1 Report

I am happy with the changes. The manuscript can now be accepted